



# Modeling the impact of chlorine emissions from coal combustion and prescribed waste incineration on tropospheric ozone formation in China

Yiming Liu[1,2], Qi Fan[1,2], Xiaoyang Chen[1,2], Jun Zhao[1,2], Zhenhao Ling[1,2], Yingying Hong[3], Weibiao Li[1,2], Xunlai Chen[4], Mingjie Wang[4], Xiaolin Wei[4]

[1]School of Atmospheric Sciences, Sun Yat-sen University, Guangzhou, 510275, China
[2]Guangdong Province Key Laboratory for Climate Change and Natural Disaster Studies, Guangzhou, 510275, China
[3]Guangdong Ecological Meteorology Center, Guangzhou, 510640, China
[4]Shenzhen Key Laboratory of Severe Weather in South China, Shenzhen, 518040, China

*Correspondence to*: Qi Fan (eesfq@mail.sysu.edu.cn), Jun Zhao (zhaojun123@mail.sysu.edu.cn)

**Abstract.** Chlorine radicals can enhance atmospheric oxidation which potentially increase tropospheric ozone concentration. However, few studies have been done to quantify the impact of chlorine emissions on ozone formation in China due to lack of chlorine emission inventory used in air quality model with sufficient resolution. In this study, Anthropogenic Chlorine Emissions Inventory for China (ACEIC) was developed for the first time, including emissions of hydrogen chloride (HCl) and molecular chlorine ($Cl_2$) from coal combustion and prescribed waste incineration (waste incineration plant). The HCl and $Cl_2$ emissions from coal combustion in China in 2012 were estimated to be 232.9 and 9.4 Gg, respectively, while HCl emission from prescribed waste incineration was estimated to be 2.9 Gg. Spatially high chlorine emissions were found in the North China Plain, the Yangtze River Delta and the Sichuan Basin. Air quality model simulations with the Community Multiscale Air Quality (CMAQ) modeling system were performed for November 2011 and the modeling results derived with and without chlorine emissions were compared. The magnitude of the simulated HCl, $Cl_2$ and $ClNO_2$ agreed reasonably with the observation when anthropogenic chlorine emissions were included in the model. The inclusion of the ACEIC increased the concentration of fine particulate $Cl^-$, leading to enhanced heterogeneous reactions between $Cl^-$ and $N_2O_5$ which resulted in the higher production of $ClNO_2$. Photolysis of $ClNO_2$ and $Cl_2$ in the morning and the reaction of HCl with OH in the afternoon produced chlorine radicals which accelerated tropospheric oxidation. When anthropogenic chlorine emissions were included in the model, the monthly mean concentrations of fine particulate $Cl^-$, daily maximum 1-h $ClNO_2$ and Cl radicals were estimated to increase by up to about 2.0 μg m$^{-3}$, 773 pptv and $1.5\times10^3$ molecule cm$^{-3}$ in China, respectively. Meanwhile, the monthly mean daily maximum 1-h $O_3$ concentration was found to increase by up to 2.2 ppbv (3.8%), while the monthly mean $NO_x$ concentration decreased by up to 0.5 ppbv (6.1%). The anthropogenic chlorine emissions potentially increased the 1-h $O_3$ concentration by up to 7.7 ppbv in China. This study highlights the need for the inclusion of anthropogenic chlorine emission on air quality modeling and demonstrated its importance on tropospheric ozone formation.



# 1 Introduction

Chlorine radicals (Cl) are highly reactive, playing a significant role in the oxidative chemistry of the troposphere (Faxon and Allen, 2013; Young et al., 2014). Similar to the hydroxyl radicals (OH), the chlorine radicals can oxidize Volatile Organic Compounds (VOCs) which potentially enhances ozone formation. In general, chlorine radicals are more reactive towards most of the VOCs than the hydroxyl radicals. The reaction rate constants of chlorine radicals with many alkanes, aromatics, alcohols and ethers range typically between one to two orders of magnitude greater than the corresponding values with the hydroxyl radicals (Aschmann and Atkinson, 1995; Nelson et al., 1990; Wang et al., 2005). Hence, the high reaction rates make the chlorine radicals competitive to the OH radicals though the concentration of chlorine radicals is an order of magnitude or more lower than that of the hydroxyl radicals (Wingenter et al., 1999). Chlorine radicals can be produced from photo-dissociation and oxidations of many of the most common chlorinated organic species, but these reaction rates are generally not fast enough to contribute significant concentrations of chlorine radicals. $ClNO_2$, $Cl_2$ and HCl are dominant primary chlorine radical sources. Riedel et al. (2012) reported that the relative contributions to Cl radicals from $ClNO_2$, $Cl_2$, and HCl were approximately 45%, 10% and 45%, respectively, in the Los Angeles regions by using a simple box model with local observation. While photolysis of $ClNO_2$ and $Cl_2$ occur in the morning, the reaction of HCl with OH occurs in the afternoon. The reactions were shown as follows:

$$ClNO_2(g) + Hv \rightarrow Cl(g) + NO_2(g) \tag{1}$$

$$Cl_2(g) + Hv \rightarrow 2Cl(g) \tag{2}$$

$$HCl(g) + OH \rightarrow Cl(g) + H_2O \tag{3}$$

In the troposphere, nitryl chloride is formed primarily by the heterogeneous reaction between $N_2O_5$ and $Cl^-$ (Eq. (4-7)) (Bertram and Thornton, 2009; Roberts et al., 2009), while the latter reactant $Cl^-$ is the major product of HCl neutralization (Seinfeld and Pandis, 1998; Pio and Harrison, 1987). Therefore, identification of emission sources and quantification of their contributions to ambient HCl levels are of critical importance for the estimation of the abundance of $ClNO_2$ and/or Cl radicals.

$$N_2O_5(g) \leftrightarrow NO_3^-(aq) + NO_2^+(aq) \tag{4}$$

$$NO_3^-(aq) + H^+(aq) \rightarrow HNO_3(g) \tag{5}$$

$$NO_2^+(aq) + H_2O \rightarrow HNO_3(g) + H^+ \tag{6}$$

$$NO_2^+(aq) + Cl^-(aq) \rightarrow ClNO_2(g) \tag{7}$$

The major sources of tropospheric HCl in the atmosphere include natural sources from sea salt (Keene et al., 1999) and biomass burning (Andreae et al., 1996), and anthropogenic sources from coal combustion and waste incineration (McCulloch et al., 1999). The global annual emission rates of HCl from sea salt and biomass burning were estimated to be 50 Tg Cl yr$^{-1}$



(Graedel and Keene, 1995; Keene et al., 1999) and 6 Tg Cl yr$^{-1}$ (Lobert et al., 1999), respectively. Although the emission rates from natural sources are much higher than the anthropogenic counterparts, they are relatively constant and well estimated. The corresponding anthropogenic emission rates from coal combustion and waste incineration were previously estimated to be 4.6 and 2 Tg Cl yr$^{-1}$, respectively (McCulloch et al., 1999). Waste incinerations include open waste incineration (the uncontrolled emissions from both residential and dump waste burning) and prescribed waste incineration (the emission from waste incineration plant). The contribution of open waste incineration to HCl was estimated to be 1 Tg yr$^{-1}$ in China (Wiedinmyer et al., 2014), while that of prescribed waste was unknown, awaiting further investigation.

Molecular Chlorine (Cl$_2$) is another important precursor of Cl radicals. However, only a few studies on its emissions are available in the literature. Chang et al. (2002) compiled an emission inventory of Cl$_2$ and HOCl for Houston and estimated an emission of about 10$^4$ kg per day in total in southeast Texas. Deng et al. (2014) collected the flue gas samples from six pulverized coal boiler units of four coal-fired power plants in China and found that about 3.6% of chlorine in coal could release in the form of gaseous Cl$_2$ during combustion. Measurements of Cl$_2$ in urban environment are also sparse, although some were made in marine air and at polar sunrise in the past (Finley et al., 2008; Lawler et al., 2011; Spicer et al., 2002; Impey et al., 1997).

Once the chlorine emission inventory was constructed, the effects of the chlorine radicals on tropospheric ozone formation can be assessed with air quality models by incorporating the emission inventory. For example, the simulation results of the comprehensive air quality model with extensions (CAMx) found that the emissions of HCl and HOCl could increase 1-h averaged O$_3$ concentration by 70 ppbv in very localized areas (Chang and Allen, 2006). Furthermore, Sarwar and Bhave (2007) estimated the effect of chlorine emission on ozone formation in the eastern United State through model simulations. They found that the monthly mean daily maximum 1-h ozone mixing ratios could be enhanced by up to 3 ppbv in Houston area when the anthropogenic emissions of Cl$_2$ and HOCl and the chlorine from sea salt aerosols were considered.

However, the role of the oxidation of hydrocarbons by Cl radicals on O$_3$ formation is still unclear in China, which is mainly due to the lack of an up-to-date anthropogenic chlorine emission. For example, the most widely used emission inventory, the Multi-resolution Emission Inventory for China (MEIC), which is developed by Tsinghua University (http://www.meicmodel.org), does not include the HCl and Cl$_2$ emissions. On the other hand, though a reactive chlorine emission inventory (RCEI) in 1990 from coal combustion and waste burning was developed by McCulloch et al. (1999), covering each country all around the world with a resolution of 1$^o$×1$^o$, it could not represent the present situation in China due to the fast industrial and economic development in recent years. Li et al. (2016) applied RCEI emission inventory in the WRF-Chem model to simulate the air quality in Pearl River Delta of China. Results from sensitivity experiments showed that the simulated particulate Cl$^-$ and ClNO$_2$ concentrations were highly sensitive to the chlorine emissions. There is hence a need to develop an up-to-date anthropogenic chlorine emission inventory in China in order to better model ClNO$_2$ production and to quantify its effect on atmospheric chemistry and air quality. Development of anthropogenic chlorine emission inventory can also help policy-makers to propose better strategies in air quality management.



In this study, Anthropogenic Chlorine Emission Inventory for China (ACEIC) was developed for the first time to include the emissions of hydrogen chloride and molecular chlorine from coal combustion and prescribed waste incineration in China. This emission inventory was then applied to the Community Multiscale Air Quality (CMAQ) modeling system to evaluate the effects of chlorine emissions on photochemical $O_3$ formation through sensitivity analysis. Simulations were performed

for November 2011 and the results derived with and without ACEIC were compared. Section 2 describes the development of chlorine emissions in China. Section 3 presents model simulation to quantify the impact of these anthropogenic chlorine emissions on atmospheric oxidation and ozone formation.

## 2 Chlorine emission inventory for China

### 2.1 Emission from coal combustion

#### 2.1.1 Coal consumption database

Coal consumption data are needed for estimating the chlorine emissions. We selected 2012 as the base year of this emission inventory. The database of coal consumptions was constructed based on the data from China Energy Statistical Yearbook (CESY, National Bureau of Statistics, 2013), which include 31 provinces, municipalities and autonomous regions. A zero value of coal consumption in Tibet was assumed in the database due to unavailable data in the yearbook. Besides, coal

combustions in Hong Kong and Taiwan were taken from International Energy Agency energy statistics (IEA, 2012) and were included in the database. Hence a total of 33 regions were included in this inventory (Table 1). Similar to the classification method used in the MEIC emission inventory, in the ACEIC we classified the coal consumption from CESY into four economic sectors according to their characteristics: (1) power plant sector, including electricity plants, heat plants and combined heat and power (CHP) plants; (2) industrial sector, including iron and steel, non-ferrous metals and other

categories bearing large-scale combustion processes; (3) residential sector, including personal consumptions in both urban and rural regions; (4) other sector, including agriculture, forestry, animal husbandry, fishery, water conservancy, construction, transport, storage, post, wholesale, retail trade, hotel, restaurants and other consumptions. Columns 2-5 in Table 1 list the coal consumptions among different categories in different provinces (or regions), along with various chlorine content in coal as discussed in the following section. In 2012, a total of 3.6 million tons of coal were consumed with

Shandong (0.3 million tons) being the highest consumer and Hainan the lowest (about 9000 tons, Tibet not included).

#### 2.1.2 Chlorine contents in coal

Chlorine is enriched in coal to some extent and is volatilized during the coal combustion process. The chlorine content in coal is essential for emission estimation and it can vary from region to region. It was reported that the chlorine content in coal in China ranged from 50 and 500 µg g$^{-1}$ with an average value of ~220 µg g$^{-1}$ (Tang and Chen, 2002), lower than most

of other countries. Meanwhile, average chlorine contents between 200 and 300 µg g$^{-1}$ in China were also reported (Lu, 1996;



Zhao et al., 1999). Chen (2010) reported a wide range of chlorine content (39-637 µg g$^{-1}$) with an average chlorine content of 280 µg g$^{-1}$, falling within those mentioned above. In this study, we chose chlorine contents reported from Chen (2010) to estimate chlorine emissions in all regions except Shanghai, Tianjin, Hong Kong and Taiwan, in which the chlorine contents were not listed. For those regions, a chlorine content of 280 µg g$^{-1}$ was assumed.

### 2.1.3 Chlorine emission factors

Chlorine emission factors from coal combustion vary with boilers and removal facilities. Table 2 summarizes the chlorine emission factors depending on the combination of boiler types and pollution control technologies in coal combustion (Jiang et al., 2005). This combination can vary significantly from one sector to another.

The net emission factors (EF) were estimated by the following Eq. (8):

$$EF_{i,j} = c_i \times \sum_k \left( R_{j,k} \times X_{j,k} \times \left(1 - \eta_{d_{j,k}}\right) \times \left(1 - \eta_{s_{j,k}}\right) \right) \tag{8}$$

where i represents the province (municipality, autonomous region); j represents the economic sector; k represents the energy allocation type (type of boiler and control device combination); c represents chlorine contents in coal; R is the chlorine release rate; X is the fraction of energy for a sector (energy allocation ratio); $\eta_d$ is the removal efficiency of dust-removal facility; and $\eta_s$ is the removal efficiency of sulfate-removal facility. The chlorine emission factors in power plant, industry, residential and others were calculated based on the parameters given in Table 1 and 2. They were then applied to estimate the HCl and Cl$_2$ emissions from coal combustion.

### 2.1.4 Development of the emission inventory

The HCl and Cl$_2$ emission (*E*) from coal combustion were estimated as follows:

$$E_{i,j} = M_{i,j} \times EF_{i,j} \times \rho \times \frac{1}{MM} \tag{9}$$

where *M* represents coal consumption; *MM* denotes the ratios of the molar mass of chlorine atom to the molecular weight of HCl and Cl$_2$ (35.5/36.5 for HCl and 1 for Cl$_2$); and $\rho$ is the chlorine proportion of HCl and Cl$_2$ in emitted flue gas. The flue gas contains chlorine species in form of particulate Cl$^-$, gaseous HCl and Cl$_2$, which were formed through chemical transformation during coal combustion. An average chlorine proportion of about 86.3% and 3.6% was reported respectively for HCl and Cl$_2$ in the flue gas samples, which were collected from six pulverized coal boiler units in four coal-fired power plants in China (Deng et al., 2014). We adopted this $\rho$ (86.3% and 3.6% respectively for HCl and Cl$_2$) when calculating HCl and Cl$_2$ emissions. Similar procedures were followed to estimate chlorine emissions in Hong Kong and Taiwan. Table 1 lists the calculated HCl and Cl$_2$ emissions from coal combustion in each region (columns 7-11 for HCl and columns 12-16 for Cl$_2$).



We employed the same resolution (0.25°×0.25°) as the one used in the MEIC for the meshed grid to develop the ACEIC. The HCl and $Cl_2$ emissions from each economic sector were spatially allocated into the center of each grid cell. To allocate the chlorine emissions from power plants, a database of the location of each point source was needed. We constructed the database following the procedures below: the chlorine emissions in each grid cell were determined using the emissions of

chlorine in the region, multiplied by a ratio of $SO_2$ emissions from power plants in MIX (An Asian anthropogenic emission inventory) in each grid cell to the total emissions of $SO_2$ in the region. MIX was developed for the year 2010 to support the Model Inter-comparison Study for Asia Phase III (MICS-Asia III) and the Task Force on Hemispheric Transport of Air Pollution (TF HTAP) (Li et al., 2015). MEIC emission was included in MIX. MIX data had 5 categories: power plant, industry, residential, transport and agriculture. In ACEIC, the locations and relative amount of chlorine emissions from

power plants were assumed to be the same as those of $SO_2$ emissions from the power plants in MIX, though this hypothesis might lead to small uncertainty. In this way, the spatial distributions of emissions of HCl and $Cl_2$ from coal combustion of power plants were then determined. Chlorine emissions from other sectors were spatially allocated based on population in 2012. The chlorine emissions in each grid cell were obtained using the chlorine emissions in the region, multiplied by the ratio of population in each grid cell to the total population in the region. The spatial distributions of HCl and $Cl_2$ emissions

from coal combustions of power plant, industry, residential, and others are shown in Figs. S1 and S2. The chlorine emission of each sector in eastern China was higher than that in western China.

## 2.2 Emission from prescribed waste incineration

### 2.2.1 Prescribed waste incineration database

Table 1 also the waste incineration from garbage disposal incinerators in each province/city from China Urban-Rural

Construction Statistical Yearbook (CURCSY, National Bureau of Statistics, 2012), which was used to estimate chlorine emissions from prescribed waste incineration. Note that the emissions of chlorine were calculated only for the regions with garbage disposal incinerators (totally 22 regions in this study). The information (location and daily capacity) of the garbage disposal incinerators was obtained from Information Platform for Municipal Solid Waste Incineration (www.waste-cwin.org).

### 2.2.2 The HCl emission factor for prescribed waste incineration

Domestic waste contains chlorine in materials such as vegetable matter, paper, plastic, dry cell batteries and salt (Lightowlers and Cape, 1988). The chlorine content of municipal waste is 0.5 wt% in average. An unabated emission factor of 2.2 g HCl kg$^{-1}$ for municipal solid waste was reported by Emmel et al. (1989), lower than that for ordinary household waste (3.5 g HCl kg$^{-1}$, Holland, 1991). We adopted the former value (2.2 g HCl kg$^{-1}$) when estimating the HCl emission from prescribed waste incineration. The $Cl_2$ emission from prescribed waste incineration was not included due to unavailable

literature data.

The net emission factor (*EF*) for prescribed waste incineration was estimated according to the following Eq. (10):



$$EF = EF_{raw} \times (1 - \eta_d) \times (1 - \eta_s) \qquad (10)$$

Where $EF_{raw}$ is the unabated emission factor (2.2 g kg$^{-1}$); and $\eta_d$, $\eta_s$ are chlorine removal efficiency of dust-removal facility and sulfate-removal facility, respectively. We assumed that the control technology of garbage disposal incinerator was similar to the coal combustion of power plant, and we hence used the average values of $\eta_s$ and $\eta_d$ data for the power plant sector in Table 2 to yield the HCl emission factor.

### 2.2.3 Development of the emission inventory

The HCl emissions (*E*) for prescribed waste incineration was estimated as follows:

$$E_i = M_i \times EF \qquad (11)$$

where *i* represents the province (municipality, autonomous region), and *M* denotes the amount of prescribed waste incineration. The estimated HCl emissions from prescribed waste incineration in each region are listed in Table 1.

The HCl emission at each garbage disposal incinerator was obtained using the emission in the region, multiplied by the ratio of daily capacity of each waste incineration plant to the total daily capacity of all plants in the region (Table 1). The HCl emissions at all the garbage disposal incinerator were then merged into 0.25°×0.25° grid cell. The result show that high emissions from prescribed waste burning could be seen around the coastal region of eastern China (Fig. S3).

### 2.3 The anthropogenic chlorine emission inventory for China

### 2.3.1 The HCl and Cl$_2$ emissions

The ACEIC developed in this study included HCl and Cl$_2$ emissions from coal combustion and HCl emissions from prescribed waste incineration. Table 1 shows the chlorine emissions of all the region in China including Hong Kong and Taiwan. The HCl and Cl$_2$ emissions from coal combustion in China in 2012 were estimated to be 232.9 and 9.4 Gg, respectively, and HCl emissions from prescribed waste burning were estimated to be 2.9 Gg. Figures 1a and b show the spatial distribution of the total HCl and Cl$_2$ emissions, respectively, where similar patterns were found, although in general the HCl emission is almost 20 times higher than that of the Cl$_2$ emission. North China Plain, YRD and Sichuan Basin contributed spatially high chlorine emissions. The highest HCl emission was found in Jiangsu, followed by Sichuan and Hebei provinces (Fig. 2). Chlorine emissions were relatively low in South China, including Guangdong, Hunan, Fujian, Jiangxi and Hainan, probably due to the low chlorine contents in coal used in those regions. The HCl emission from industry contributed to as high as 68% of the total emissions, followed by others (12%), residential (10%), power plant (9%) and prescribed waste incineration (1%).





### 2.3.2 Comparison with other chlorine emission

RCEI developed by McCulloch et al. (1999) was the only emission inventory for chlorine that include China, containing globally the chlorine emitted from coal combustion and waste incineration in 1990. The ACEIC developed in this study made four progresses based on the RCEI: (1) More comprehensive database of coal combustion and prescribed waste

incineration in China. The data in each province/city in China was taken from CESY and CURCSY in this study, which were more detailed than those in RCEI that was from IEA energy statistics and only included the total amount of coal consumption in China. (2) Higher spatial resolution. The ACEIC has a higher resolution (0.25°×0.25°) than the RCEI (1°×1°), providing a higher resolution for regional air quality modeling; (3) More emission factors. When estimating emission factors, the ACEIC included the removal rates of chlorine from dust-removal facility and sulfate-removal facility in China, while the

RCEI did not, leading to higher estimated HCl emission in RCEI. We estimated about 232.9 Gg HCl emission in China in 2012 in ACEIC, only about one fourth of that estimated from RCEI (866.7 Gg). (4) Accounting for $Cl_2$ emission in the inventory. The ACEIC includes $Cl_2$ emission which is also emitted during coal combustion in China based on the measurement by Deng et al. (2014).

In the following section, the ACEIC was incorporated into the CMAQ model to simulate the air quality in central and eastern

China. It was evaluated by comparing the simulated and observed concentrations of chlorine species. In addition, the effect of chlorine emissions on tropospheric ozone formation was quantified to assess its importance in atmospheric chemistry in China. The refined and updated anthropogenic chlorine emission will help to evaluate the impact of chlorine emission on ozone formation in China.

## 3 Impact of chlorine emissions on tropospheric ozone formation

### 3.1 Model setting

CMAQ was developed by United States Environmental Protection Agency (US EPA) to approach air quality as a whole by including state-of-the-science capabilities for modeling multiple air quality issues, including tropospheric ozone, fine particles, toxics, acid deposition and visibility degradation (Byun and Schere, 2006). The latest version (5.1) was used in this study. Meteorological inputs were driven by the Weather Research and Forecasting (WRF) model. The meteorological

boundary conditions and initial conditions of WRF were provided by NCEP/NCAR final (FNL) reanalysis data (1°×1°). The modeling domain with 27 km horizontal resolution is shown in Fig. 3. The number of modeled layers was 40 and the highest layer can reach the top of 50 hPa. The CMAQ modeling domain covered the central and eastern China, which was smaller than the WRF modeling domain to reduce the effect of meteorological boundary from the WRF model. The meteorology-chemistry interface processor (MCIP) was used to convert WRF outputs to CMAQ input format. The boundary conditions of

chemical species for CMAQ were provided by the Model for Ozone and Related Chemical Tracers, version 4 (MOZART-4) results (http://www.acom.ucar.edu/wrf-chem/mozart.shtml).





In this study, anthropogenic and biogenic emissions were both included in the simulation. MIX emission inventory (Li et al., 2015) was used in the simulation. International shipping emission was taken from the Hemispheric Transport Atmospheric Pollution (HTAP) emissions version 2.0 dataset (Janssens-Maenhout et al., 2015). Biogenic emission was calculated from the Model of Emissions of Gas and Aerosols from Nature (MEGAN) (Guenther et al., 2006). Besides, sea salt emission was

calculated inline during the simulation in the CMAQ model. The methods for estimating sea salt emission and its impact on aerosol chemical formation could be found in Liu et al. (2015). SAPRC07TIC mechanism (Carter, 2010; Hutzell et al., 2012; Xie et al., 2013; Lin et al., 2013; Pye et al., 2015) was selected as the gas-phase chemical mechanism in the CMAQ model. ISORROPIA (Fountoukis and Nenes, 2007) was used to model chemistry of inorganic aerosols. Detailed chlorine chemistry (including Eq. (1-7)) was considered in the CMAQ model.

The simulation was performed for November 2011. The spin-up time was 10 days (October 22−31) prior to November 2011. During the simulation period, China was controlled by high-pressure systems most of time, which hindered the transport and diffusion of air pollutants.

Two experiments were set up to evaluate the impact of chlorine emission on tropospheric ozone formation. One experiment included the HCl and $Cl_2$ emission from coal combustion and prescribed waste incineration (ACEIC) in the model (Base

experiment), while the other experiment did not (NoCl experiment). The comparison of the Base and NoCl experiments could help to quantify the impacts of anthropogenic chlorine emissions. Four typical sites in four different regions were selected to analyze the diurnal variations of chlorine species: Beijing (BJ), Shanghai (SH), Guangzhou (GZ) and Chongqing (CQ). The locations of these sites are shown in Fig. 3.

### 3.2 Evaluation of chlorine species

### 3.2.1 HCl evaluation

Table 3 presents the comparison of modeled HCl concentrations to the observed values in China from available literatures. It should be noted that the modeled and observed HCl concentrations were not pared in time and space. The modeled HCl concentrations in the Base and NoCl experiments in Beijing and Guangzhou were underestimated, while those in Shanghai and Hong Kong were overestimated. However, the modeled HCl concentrations from both experiments reasonably matched

the observations in a similar magnitude. The difference between the modeled and observed HCl concentration in Beijing reduced when the ACEIC was incorporated into the model, implying the importance of anthropogenic emissions in this region.

### 3.2.2 $Cl_2$ evaluation

Reports on $Cl_2$ measurements in the atmosphere are sparse in the literature. The $Cl_2$ concentration was measured to be about

2.3 pptv in average in La Jolla (Finley and Saltzman, 2008) and 2.5-20 pptv with a 2-month mean of 3.5 pptv in Irvine, California. We estimated a monthly average concentration of 1-10 pptv (most of urban regions) in China in this study by



incorporating the ACEIC into CMAQ system for air quality modeling (Fig. 6a), which was reasonable compared to the observed values in northern America. Cl$_2$ concentration was very low when anthropogenic chlorine emission was not included in the model. (Fig. S8)

### 3.2.3 ClNO$_2$ evaluation

Highest ClNO$_2$ concentrations in China were observed throughout the Northern Hemisphere in the CMAQ simulation (Sarwar et al., 2014). Up to about 2000 pptv ClNO$_2$ concentration was measured in Hong Kong during the summer of 2012 (Tham et al., 2014) and in Tianjin during the summer of 2014 (Tham et al., 2016). We estimated a monthly average concentration of up to 1178 pptv ClNO$_2$ in China (Fig. 4g), which was comparable to the observed values.

### 3.3 Impact of chlorine emission on atmospheric oxidation

**3.3.1 Impact of HCl emission**

Figure 4a shows the spatial distribution of monthly average HCl concentration in the Base experiment. The HCl concentration over the ocean was higher than that over the land, due probably to the largest proportion of HCl emission from the dechlorination of sea salt aerosols (Graedel and Keene, 1995; Keene et al., 1999). The highest concentration of HCl was found in South China Sea where sea salt emission was also high due to high wind speed and downwind location (Fig. S4).

The impact of chlorine emissions on HCl concentration is shown in Figs. 4b and c. The inclusion of the ACEIC in the model increased the HCl concentration by up to 1.7 μg m$^{-3}$ in inland China. The chlorine emissions accounted for up to 85.6% of the HCl concentration in Sichuan Basin (Fig. 4c). The dechlorination of sea salt aerosols transported to inland area were also considered as an important proportion of HCl concentration, especially in South China and the coastal regions in East China (Fig. S5), where the impact of anthropogenic chlorine emission was low.

The spatial distribution of monthly mean concentrations of fine particulate Cl$^-$ is shown in Fig. 4d. Higher concentration was found in North China Plain and South China Sea. The concentrations of fine particulate Cl$^-$ increased by up to 2.0 μg m$^{-3}$ when anthropogenic chlorine emissions were included in the model (Fig. 4e). The increase of fine Cl$^-$ concentration was attributed to the gas-particle HCl transformation and was sensitive to chlorine emissions, especially in Sichuan Basin, contributing up to 89% increase (Fig. 4f).

The HCl concentrations were found to significantly increase in regions such as Sichuan Basin and YRD, consistent with the high anthropogenic chlorine emissions shown in Fig. 1a. However, the increase of HCl concentration was negligible in North China Plain even though there was also high HCl emission, while surprisingly the concentration of particulate Cl$^-$ increased more significantly than that in Sichuan and YRD. Chlorine partitioning between gas and particle phases ([Cl$^-$]/([Cl$^-$]+[HCl])) was calculated and shown in Fig. 5. Higher chlorine partitioning meant that more HCl was transferred into particulate Cl$^-$.

The spatial distribution of chlorine partitioning in the Base and NoCl experiment was almost the same, suggesting that the chlorine emissions had little impact on the rate of gas-particle conversion. Higher chlorine partitioning rates were found in



North China Plain than that in other regions in inland China, where $NH_3$ emission was high (Fig. S6), leading to significant increase of particulate $Cl^-$ concentration when the ACEIC was included in the model. Meanwhile, high $NH_3$ emission in North China Plain also accelerated the gas-particle transformation from $NH_3$ to particulate $NH_4^+$, leading to the decrease of $NH_3$ concentration and the increase of $NH_4^+$ concentration in $PM_{2.5}$ (up to -1.1 µg m$^{-3}$ and 1.0 µg m$^{-3}$, respectively) (Fig. S7).

The spatial distribution of daily maximum 1-h $ClNO_2$ concentration is shown in Fig. 4g. The $ClNO_2$ concentration in North China Plain, Sichuan Basin, and the coastline along South China were significantly higher than those in other regions. The reservoir species $ClNO_2$ was formed through heterogeneous reaction between $Cl^-$ and $N_2O_5$ on the aerosol surfaces. High $Cl^-$ concentrations would accelerate the heterogeneous reaction rates, leading to enhanced $ClNO_2$ production. The inclusion of the ACEIC in the model increased the monthly daily maximum 1-h $ClNO_2$ concentration by up to 773 pptv in the whole

domain, especially in the North China Plain and Sichuan Basin (Fig. 4h), highlighting the importance of anthropogenic chlorine emissions on $ClNO_2$ formation (up to 78.4% of $ClNO_2$ production was related to anthropogenic chlorine emissions) (Fig. 4i).

### 3.3.2 Impact of $Cl_2$ emission

The spatial distribution of the monthly mean $Cl_2$ concentration is presented in Fig. 6a. As expected, high concentrations were

found in regions with high $Cl_2$ emissions, including Sichuan Basin, YRD and North China Plain (Fig. 1b). The $Cl_2$ concentration was very low ($< 3.4\times10^{-3}$ pptv) when anthropogenic chlorine emissions were not included in the model (Fig. S8). The differences between Base and NoCl experiments (Fig. 6b) showed that $Cl_2$ was almost from direct emissions, nearly 100% of $Cl_2$ was originated from inland anthropogenic chlorine emissions (Fig. 6c). The results suggested that anthropogenic chlorine emission was a significant source of $Cl_2$ which should be included in air quality modeling in order to

accurately model regional air quality.

### 3.3.3 Impact on Cl radicals

Both $ClNO_2$ (mainly from heterogeneous reaction between particulate $Cl^-$ and $N_2O_5$) and $Cl_2$ (mainly from direct emissions) can photolyze to produce Cl radicals after sunrise (Eq. (1) and (2)). The diurnal variation of $ClNO_2$, $Cl_2$ and Cl radicals are presented in Fig. 7. The $ClNO_2$ concentration continued to drop and reached a minimal value between 12 and 4 pm but

gradually increased after sunset due to the cease of photolysis and continuous accumulation of $ClNO_2$ from the heterogeneous reaction and then reached a peak just before sunrise. The $Cl_2$ concentration reached a peak at about 8 am and subsequently dropped substantially in the whole morning and early afternoon until 4 pm due to apparently photolysis. The $Cl_2$ concentration increased gradually after 4 pm and continued to accumulate during nighttime. The Cl radical concentration was peaked in the morning due to the photolysis of $Cl_2$ and $ClNO_2$, while it was mainly from the reaction of HCl with OH in

the afternoon when $Cl_2$ and $ClNO_2$ concentrations were very low. Similar diurnal cycles can be found at all sites, however, the impact of anthropogenic chlorine emissions at each site varied. Guangzhou and Shanghai site could represent as the coastal region, where the predominant sources of chlorine were from sea salt emission, while Chongqing site could represent



as the inland region, where the predominant sources of chlorine were from coal combustion and waste incineration. The impact of anthropogenic chlorine emissions in Chongqing was higher than those in other regions.

The spatial distribution of Cl radical concentration is shown in Fig. 6d. Higher Cl radical concentration was found in South China Sea, where high HCl concentration was found (Fig. S4a). The concentration of Cl radicals over the land reached up to $8 \times 10^3$ molecule cm$^{-3}$. The Cl concentrations increased by up to $1.5 \times 10^3$ molecule cm$^{-3}$ in the whole domain when anthropogenic chlorine emissions were considered in the model (Fig. 6e). The chlorine emission contributed up to 83.3% of Cl concentration (Fig. 6f).

## 3.4 Impact of chlorine emissions on tropospheric ozone formation

Atmospheric oxidation of VOCs initiated by the Cl radicals plays an important role in tropospheric ozone formation. The oxidation reaction was accelerated as the Cl concentration increased and hence may have potential impacts on $NO_x$ chemistry and ozone formation. The $NO_x$ concentration was observed to be high in North China Plain (Fig. 8a). It decreased by up to 0.5 ppbv (6.1%) when the anthropogenic chlorine emissions were included in the model (Figs. 8b-c). In particular, region such as North China Plain and Sichuan Basin was significantly affected by the chlorine emissions. The $O_3$ concentration was high in South China, Sichuan Bain and southwestern China during November 2011 and increased by up to 2.2 ppbv (3.8%) when anthropogenic chlorine emission was included (Figs. 8d-f), especially in central China. The impact of ACEIC was reasonable compared to the result reported in Houston area (up to 3 ppbv increase) by Sarwar and Bhave (2007). The $O_3$ concentration increased corresponding to the decrease of $NO_x$ concentration (Figs. 8b and 8e). This is attributed to the facts that more $NO_x$ conversion leads to more ozone production when chlorine emissions were included in the model. The maximum impact of chlorine emissions on 1-h $O_3$ concentration is shown in Fig. 9. The largest increase of 1-h $O_3$ concentration was found along the Yangtze River, where the chlorine emission potentially increased the 1-h $O_3$ concentration by up to 7.7 ppbv.

## 4 Conclusions

The Anthropogenic Chlorine Emission Inventory for China (ACEIC) was developed for the first time, which included HCl and $Cl_2$ from coal combustion and prescribed waste incineration. The HCl and $Cl_2$ emissions from coal combustion in China in 2012 were estimated to be 232.9 Gg and 9.4 Gg respectively, while HCl emission from prescribed waste incineration in China was estimated to be 2.9 Gg. The high chlorine emission was found in North China Plain, Yangtze River Delta and Sichuan Basin. In ACEIC, HCl emissions from coal combustion of industry contributed 68% of the total emission, followed by others, residential, power plant and prescribed waste incineration.

The modeling results with ACEIC showed that the simulated HCl, $Cl_2$ and $ClNO_2$ agreed reasonably with the observed values. The inclusion of anthropogenic chlorine emissions in the model increased the concentration of fine particulate $Cl^-$, leading to enhanced heterogeneous reaction of $Cl^-$ with $N_2O_5$ which produced $ClNO_2$. Reaction of HCl with OH and



photolysis of $ClNO_2$ and $Cl_2$ produce chlorine radicals. The monthly mean concentrations of fine particulate $Cl^-$, daily maximum 1-h $ClNO_2$, Cl radicals increased by up to 2.0 $\mu g\ m^{-3}$, 773 pptv, and $1.5 \times 10^3$ molecule $cm^{-3}$ when anthropogenic chlorine emission was included in the model. In inland China, up to 89%, 78.4% and 83.3% of monthly mean concentrations of fine particulate $Cl^-$, daily maximum 1-h $ClNO_2$, Cl radicals came from anthropogenic chlorine emissions, respectively.

The Cl radicals reacted with VOCs and potentially enhanced $O_3$ concentration. The monthly mean concentration of daily 1-h maximum $O_3$ increased by up to 2.2 ppbv (3.8%) when the ACEIC was included in the model. The chlorine emission potentially increased the 1-h $O_3$ concentration by up to 7.7 ppbv in China. As the precursor of $O_3$, the monthly mean concentration of $NO_x$ decreased by up to 0.5 ppbv (6.1%). Significant increase of daily maximum 1-h $O_3$ was found in the central China, corresponding to the region with significant decrease of $NO_x$.

More attention should be paid to the influence of chlorine emissions. The impact of chlorine emissions on ozone formation might vary from season to season. In the future, other typical months will be simulated and analyzed. In addition, emissions of hydrogen chloride and molecular chlorine not only help the increase of tropospheric ozone concentration, but also enhance the concentrations of particulate $NH_4^+$. Further studies should focus on the impact of chlorine emissions on secondary aerosol formation and deposition.

**Acknowledgements**

This work was supported by the National Key Research and Development Program of China (2017YFC0210105, 2016YFC0202206); the National Natural Science Foundation of China (NSFC) (91544102, 21577177); the Science and Technology Planning Project of Guangdong Province, China (2014B020216003, 2016B050502005, 2014A020216008); the Science and Technology Planning Project of China (2014BAC21B02); and the National Key Research and Development

Program of China (2016YFC0203600). This work was also partly supported by the Jiangsu Collaborative Innovation Center for Climate Change and the high-performance grid-computing platform of Sun Yat-sen University. The authors acknowledge Professor Qiang Zhang of Tsinghua University for sharing the MIX inventory.

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





**Table 1: Coal consumption and emissions of hydrogen chlorine and molecular chlorine, listed alphabetically according to regions.**

| Region | Coal consumption (Gg) | | | | Chlorine content in coal (µg g⁻¹) | HCl emission from coal consumption (Mg) | | | | | Cl₂ emission from coal consumption (Mg) | | | | | Waste incineration (Gg) | HCl emission from waste incineration (Mg) |
|---|---|---|---|---|---|---|---|---|---|---|---|---|---|---|---|---|---|
| | Power | Industry | Residential | Other | | Power | Industry | Residential | Other | Total | Power | Industry | Residential | Other | Total | | |
| Anhui | 108519 | 30508 | 530 | 628 | 132 | 479 | 2213 | 58 | 73 | 2823 | 19 | 89 | 2 | 3 | 114 | 1159 | 86 |
| Beijing | 12574 | 3967 | 2722 | 3446 | 90 | 38 | 196 | 204 | 272 | 711 | 2 | 8 | 8 | 11 | 29 | 947 | 70 |
| Chongqing | 17806 | 32342 | 1941 | 5142 | 617 | 369 | 10998 | 999 | 2788 | 15154 | 15 | 444 | 40 | 113 | 612 | 950 | 70 |
| Fujian | 47660 | 32767 | 790 | 987 | 39 | 62 | 703 | 26 | 34 | 824 | 3 | 28 | 1 | 1 | 33 | 2501 | 185 |
| Gansu | 38760 | 14486 | 5157 | 1891 | 250 | 325 | 1992 | 1074 | 415 | 3805 | 13 | 80 | 43 | 17 | 154 | | |
| Guangdong | 118779 | 53382 | 636 | 1071 | 67 | 266 | 1962 | 35 | 63 | 2325 | 11 | 79 | 1 | 3 | 94 | 4947 | 367 |
| Guangxi | 32622 | 31182 | 368 | 1188 | 270 | 296 | 4644 | 83 | 282 | 5305 | 12 | 187 | 3 | 11 | 214 | 269 | 20 |
| Guizhou | 54049 | 35384 | 8772 | 13929 | 165 | 299 | 3214 | 1206 | 2017 | 6736 | 12 | 130 | 49 | 81 | 272 | | |
| Hainan | 6914 | 2392 | 0 | 0 | 67 | 15 | 88 | 0 | 0 | 103 | 1 | 4 | 0 | 0 | 4 | 616 | 46 |
| Hebei | 111357 | 74229 | 13945 | 7355 | 310 | 1157 | 12671 | 3603 | 2002 | 19433 | 47 | 512 | 145 | 81 | 785 | 1255 | 93 |
| Heilongjiang | 64369 | 24149 | 4037 | 12274 | 194 | 418 | 2577 | 652 | 2088 | 5735 | 17 | 104 | 26 | 84 | 232 | 92 | 7 |
| Henan | 126359 | 59252 | 11380 | 2084 | 263 | 1114 | 8579 | 2494 | 481 | 12669 | 45 | 346 | 101 | 19 | 512 | 910 | 67 |
| Hubei | 38967 | 85429 | 4828 | 15521 | 90 | 117 | 4216 | 361 | 1221 | 5914 | 5 | 170 | 15 | 49 | 239 | 2102 | 156 |
| Hunan | 36273 | 53428 | 5231 | 10477 | 61 | 75 | 1807 | 268 | 565 | 2714 | 3 | 73 | 11 | 23 | 110 | 227 | 17 |
| Inner Mongolia | 225139 | 29432 | 17350 | 26335 | 165 | 1246 | 2675 | 2387 | 3816 | 10124 | 50 | 108 | 96 | 154 | 409 | | |
| Jiangsu | 189947 | 55263 | 141 | 626 | 637 | 4062 | 19409 | 75 | 350 | 23896 | 164 | 784 | 3 | 14 | 965 | 7077 | 525 |
| Jiangxi | 26976 | 25154 | 1350 | 587 | 76 | 69 | 1051 | 85 | 39 | 1244 | 3 | 42 | 3 | 2 | 50 | | |
| Jilin | 95761 | 38895 | 3948 | 2422 | 187 | 601 | 4007 | 616 | 398 | 5621 | 24 | 162 | 25 | 16 | 227 | 564 | 42 |
| Liaoning | 95761 | 38895 | 3948 | 2422 | 546 | 1755 | 11707 | 1799 | 1162 | 16424 | 71 | 473 | 73 | 47 | 663 | 296 | 22 |
| Ningxia | 53381 | 11525 | 662 | 592 | 209 | 375 | 1328 | 116 | 109 | 1927 | 15 | 54 | 5 | 4 | 78 | | |
| Qinghai | 6166 | 4531 | 1076 | 780 | 170 | 35 | 425 | 153 | 117 | 729 | 1 | 17 | 6 | 5 | 29 | | |
| Shaanxi | 60902 | 33625 | 4311 | 4933 | 194 | 396 | 3588 | 696 | 839 | 5519 | 16 | 145 | 28 | 34 | 223 | | |
| Shandong | 199887 | 115266 | 4654 | 18305 | 180 | 1208 | 11436 | 699 | 2895 | 16237 | 49 | 462 | 28 | 117 | 656 | 3196 | 237 |
| Shanghai | 37199 | 8358 | 412 | 982 | 280 | 350 | 1290 | 96 | 242 | 1977 | 14 | 52 | 4 | 10 | 80 | 1036 | 77 |
| Shanxi | 125466 | 43747 | 12454 | 8345 | 366 | 1540 | 8818 | 3800 | 2682 | 16839 | 62 | 356 | 153 | 108 | 680 | 1092 | 81 |
| Sichuan | 27899 | 60105 | 3402 | 784 | 581 | 544 | 19252 | 1650 | 401 | 21846 | 22 | 777 | 67 | 16 | 882 | 460 | 34 |
| Tianjin | 37627 | 9701 | 676 | 1705 | 280 | 354 | 1497 | 158 | 419 | 2428 | 14 | 60 | 6 | 17 | 98 | 823 | 61 |
| Tibet | 0 | 0 | 0 | 0 | | 0 | 0 | 0 | 0 | 0 | 0 | 0 | 0 | 0 | 0 | | |
| Xinjiang | 61655 | 26070 | 2550 | 2680 | 262 | 543 | 3770 | 558 | 618 | 5490 | 22 | 152 | 23 | 25 | 222 | | |
| Yunnan | 31698 | 28526 | 3929 | 3834 | 199 | 211 | 3122 | 651 | 669 | 4654 | 9 | 126 | 26 | 27 | 188 | 1480 | 110 |
| Zhejiang | 108519 | 30508 | 530 | 628 | 637 | 2321 | 10715 | 282 | 352 | 13669 | 94 | 433 | 11 | 14 | 552 | 6768 | 502 |
| Mainland China | 2198991 | 1092498 | 121730 | 151953 | | 20640 | 159950 | 24884 | 27409 | 232875 | 835 | 6457 | 1002 | 1106 | 9406 | 38764 | 2874 |
| Hong Kong | 10126 | 2225 | 0 | 0 | 280 | 95 | 343 | 0 | 0 | 439 | 4 | 14 | 0 | 0 | 18 | | |
| Taiwan | 46731 | 9243 | 0 | 37 | 280 | 439 | 1426 | 0 | 9 | 1875 | 18 | 58 | 0 | 0 | 76 | | |





**Table 2: Emission factors of chlorine from coal combustion in China**

| Economic Sector | Boiler type | Pollution control technology | Energy allocation ratio (%)[a] | Chlorine release rate (%) | Removal efficiency from dust-removal facility (%) | Removal efficiency from sulfate-removal facility (%) |
|---|---|---|---|---|---|---|
| Power plant | Pulverized coal boiler | Cottrell | 43 | 98.5[b] | 5.1[b] | 95.5[b] |
| | Pulverized coal boiler | Bag-type dust remover | 43 | 98.5[b] | 10.4[b] | 95.5[b] |
| | Pulverized coal boiler | Wet-type dust remover | 6 | 98.5[b] | 60.0[c] | 95.5[b] |
| | Grate furnace | Wet-type dust remover | 7 | 99[e] | 60.0[c] | 95.5[b] |
| | Grate furnace | Mechanical dust collector | 1 | 99[e] | 25[f] | 95.5[b] |
| Industry | Grate furnace | Wet-type dust remover | 29 | 99[e] | 60.0[c] | 0 |
| | Grate furnace | Mechanical dust collector | 58 | 99[e] | 25[f] | 0 |
| | Grate furnace | No | 4 | 99[e] | 0 | 0 |
| | Fluidized bed boiler | Wet-type dust remover | 9 | 99.6[d] | 60.0[c] | 0 |
| Residential | Traditional stove | No | 19 | 94[g] | 0 | 0 |
| | Reinforced stove | No | 41 | 94[g] | 0 | 0 |
| | Tea bath | No | 4 | 94[g] | 0 | 0 |
| Other | Grate furnace | No | 100 | 99[e] | 0 | 0 |

[a] From Jiang et al. (2005); [b] from Deng et al. (2014); [c] from Jiang et al. (2004); [d] from Lopez-Vilarino et al. (2003); [e] from Meij (1991); [f] from Mei et al. (2006); [g] Iapalucci et al. (1969).





**Table 3: A comparison of predicted HCl concentrations for the Base and NoCl experiments to observed data from literature. [a]**

| Location | Period | Observation | Base | NoCl | Reference |
|---|---|---|---|---|---|
| Beijing, China | Winter 2007 | 0.22 | | | Ianniello et al. (2011) |
| Beijing, China | Summer 2007 | 0.45 | 0.12 | 0.06 | Ianniello et al. (2011) |
| Beijing, China | Jul and Aug 2002 and 2003 | 0.6 | | | Wu et al. (2009) |
| Beijing, China | Jul-Aug 2001 | 0.3-0.8 | | | Yao et al. (2003) |
| Shanghai, China | Oct-Nov 2012 | 0.5 | 0.87 | 0.64 | Shi et al. (2014) |
| Guangzhou, China | Oct-Nov 2004 | 2.8 | 1.10 | 1.05 | Hu et al. (2008) |
| Hong Kong | Autumn 2000 | 0.8 | 1.27 | 1.18 | Yao et al. (2006) |

[a] Units are $\mu g\ m^{-3}$. Note that the observed and model values are not paired in time and space. Model predictions are taken from the general geographic areas of the observed data.

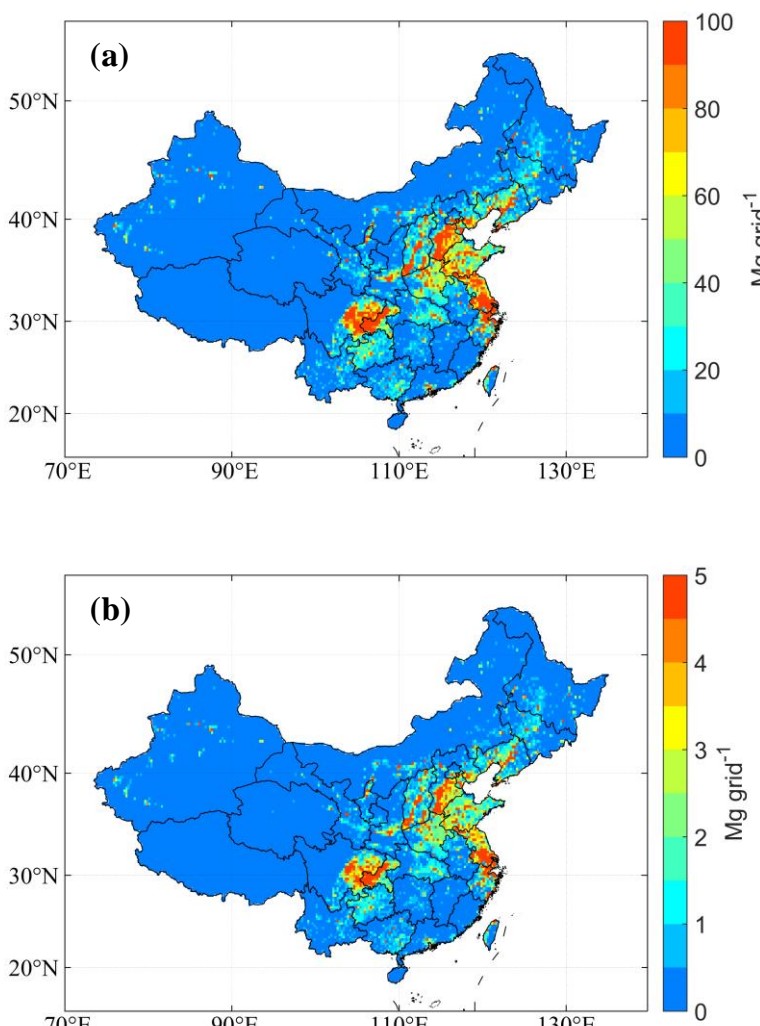

**Figure 1: Spatial distribution of the emissions of hydrogen chloride (a) and molecular chlorine (b) in the ACEIC.**




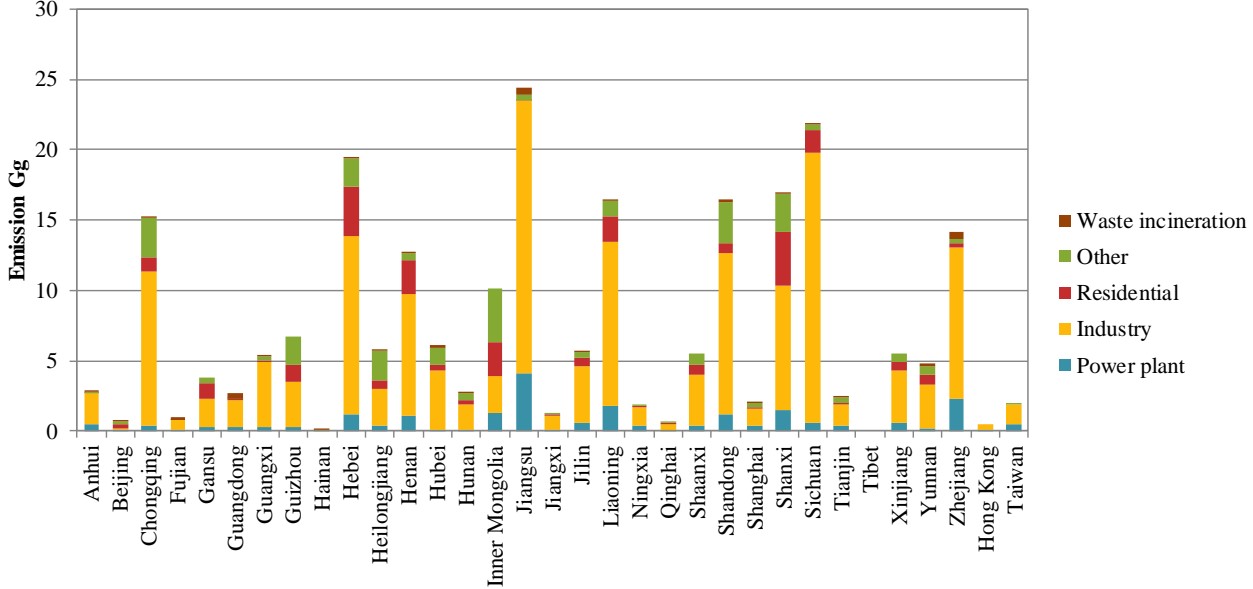

**Figure 2: The HCl emission in China in 2012 from four economic sectors of coal combustion and from prescribed waste incineration.**





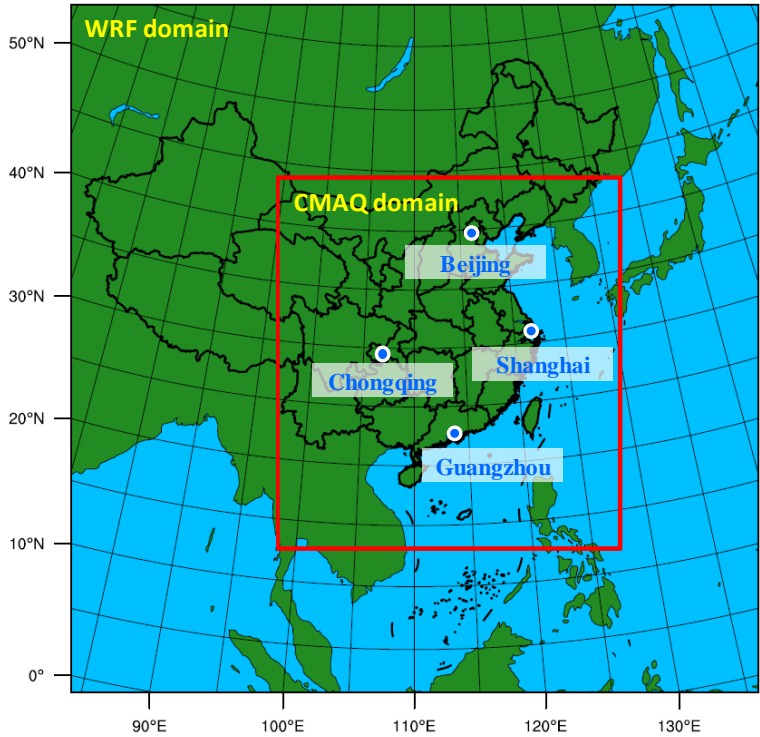

**Figure 3: Modeling domain of WRF/CMAQ and the locations of typical sites: Beijing, Shanghai, Guangzhou and Chongqing.**





| | Base | Base-NoCl | (Base-NoCl)/Base×100% |
|---|---|---|---|
| HCl | (a) Min=0, Max=2.5 | (b) Min=0, Max=1.7 | (c) Min=-0.4, Max=85.6 |
| Fine particulate Cl⁻ | (d) Min=0, Max=2.7 | (e) Min=0, Max=2 | (f) Min=-0.2, Max=89 |
| Daily maximum 1-h ClNO₂ | (g) Min=0, Max=1178.4 | (h) Min=-3.2, Max=773 | (i) Min=-4.2, Max=78.4 |

**Figure 4: Comparison of the monthly mean concentrations of HCl, fine particulate Cl⁻ and daily maximum 1-h ClNO₂ in the Base experiment, the differences (Base minus NoCl), and the fraction change to the Base experiment.**



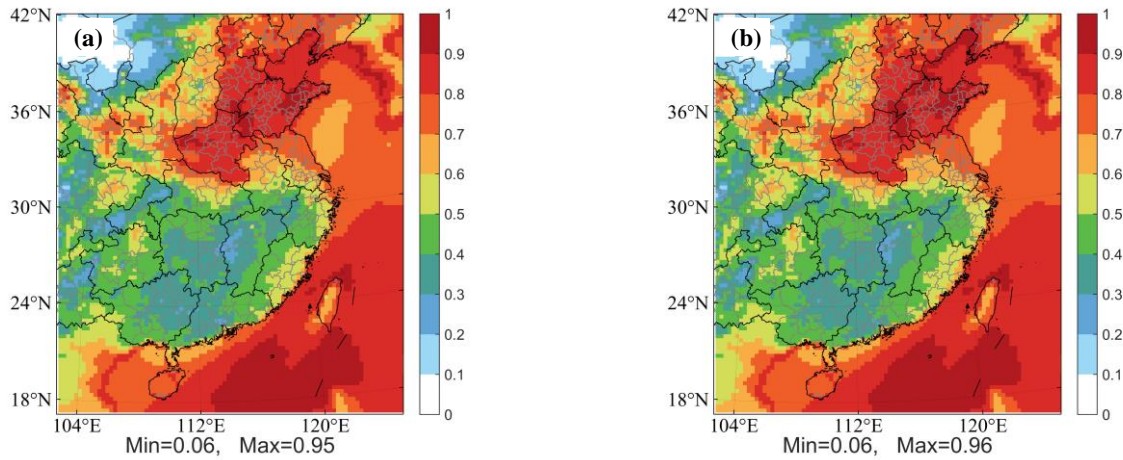

**Figure 5: Spatial distribution of the monthly mean of chlorine partitioning ([Cl-]/([Cl-]+[HCl])) in the Base (a) and NoCl (b) experiment.**





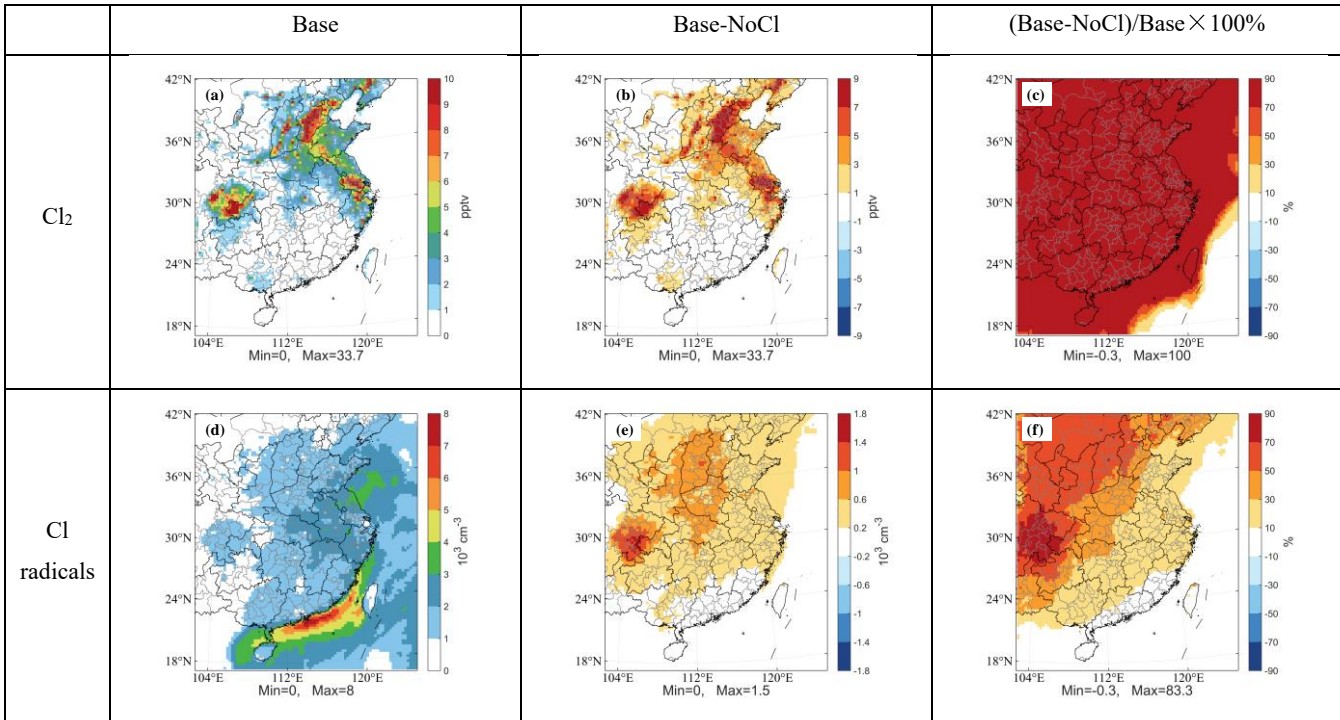

**Figure 6: Same as Fig. 4 but for Cl₂ and Cl radicals.**





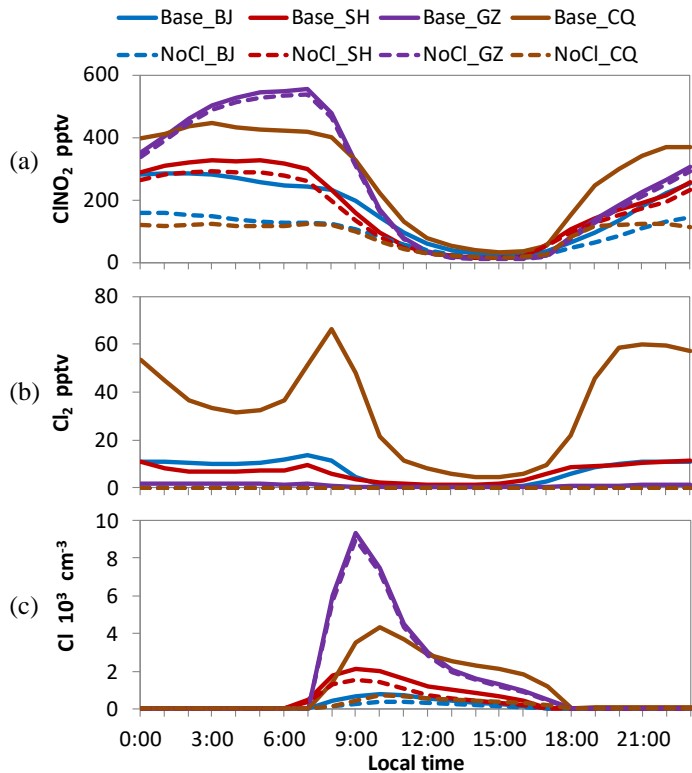

**Figure 7: Diurnal variations of monthly mean concentrations of $ClNO_2$ (a), $Cl_2$ (b) and Cl radical (c) in Beijing, Shanghai, Guangzhou and Chongqing in the Base and NoCl experiments.**





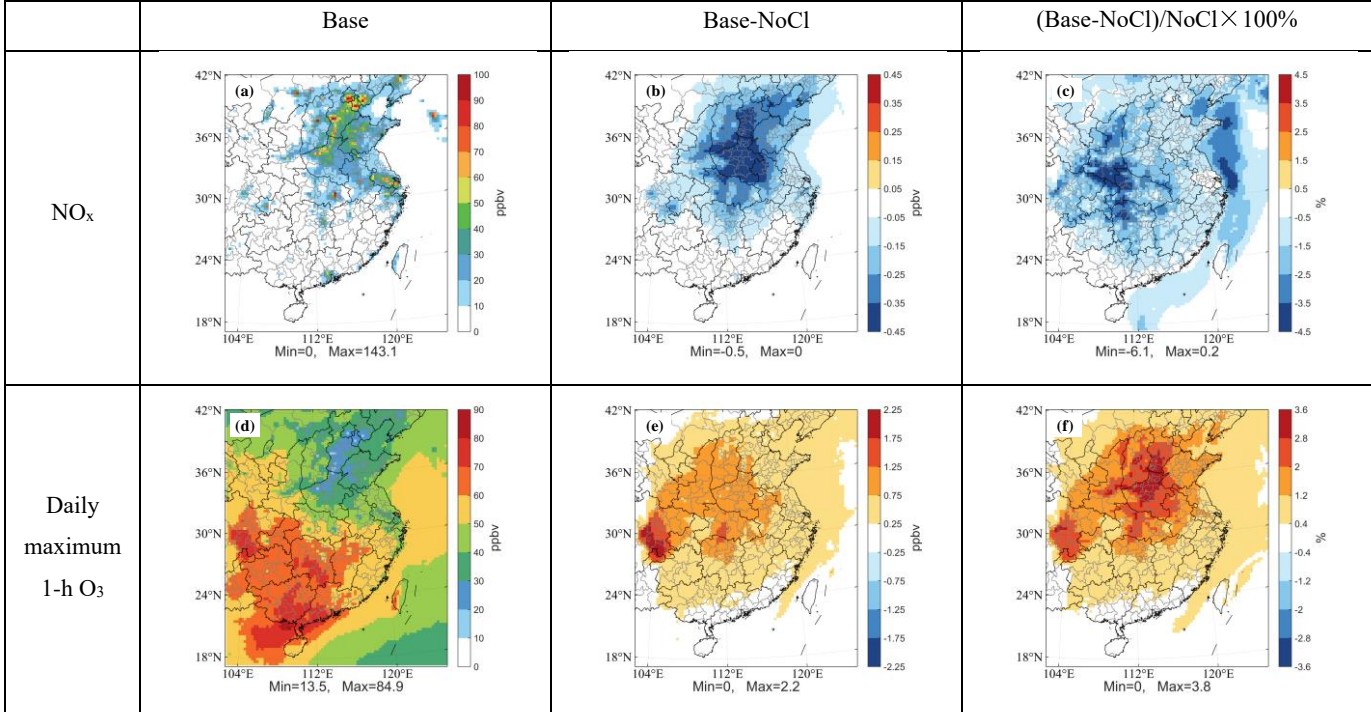

**Figure 8: Comparison of the monthly mean concentrations of NOₓ and daily maximum 1-h O₃ in the Base experiment, the differences (Base minus NoCl), and the fraction change to the NoCl experiment.**

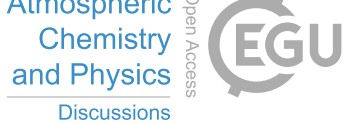



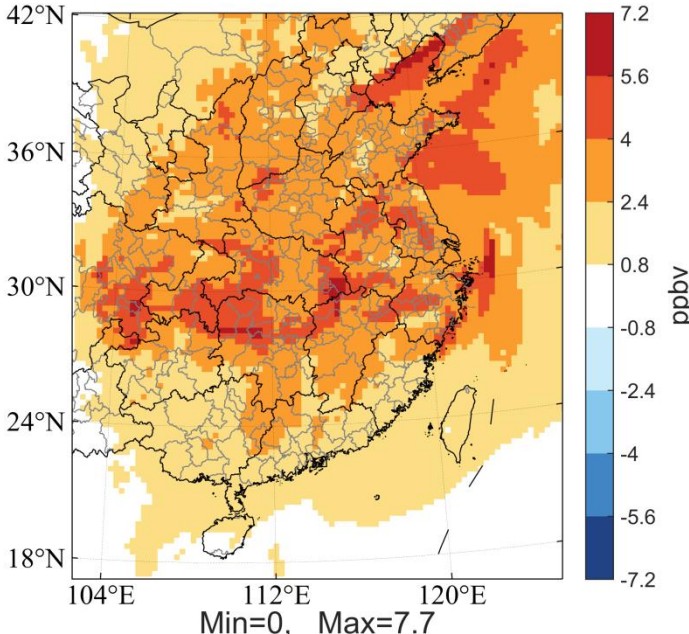

**Figure 9: Spatial distribution of the maximum impact of chlorine emissions on 1-h O₃ concentration in November 2011.**

