# Peer review of "Modeling the impact of chlorine emissions from coal combustion and prescribed waste incineration on tropospheric ozone formation in China"

_Atmospheric Chemistry and Physics, 2017_

## Referee Comment (RC1) · Anonymous Referee #2 · 29 Nov 2017

General comments The authors estimated chlorine emissions from coal combustion and prescribed waste incineration in China and used the CMAQ model to examine the impact of these chlorine emissions on ozone. Overall, the article is written clearly and merits publication. However, several issues need to be addressed before publication.

Specific comments

Page 2, line 16-17, equation 1-2 The authors may replace "H$\gamma$" in both reactions with "h$\gamma$".

Page 6, line 19-21 Incomplete sentence

[Figure]

Page 7, line 19-29 How the chlorine emissions were temporally allocated?

Page 9, line 13-15 ACEIC has been defined before. It appears that ACEIC has been defined several times throughout the article. Please check and define it once.

Page 11, line 1-10 The authors reported that chlorine emissions/chemistry increased the conversion of NH3 into NH4+. What mechanism caused the increased conversion of NH3 into NH4+?

Page 12, line 9-21 The authors demonstrated the impact of chlorine emissions/chemistry on daily maximum 1-hr O3. Chlorine emissions/chemistry tends to increase ozone in the morning hours. Thus, it is likely to have a larger impact on 8-hr O3 on than on 1-hr O3. It will be important to present impact of chlorine emissions/chemistry on 8-hr O3 also.

Page 13, line 1-10 The authors reported that chlorine emissions/chemistry decreased NOx concentrations without providing any explanation? Why NOx concentration decreased?

Figure 1 Unit of chlorine emissions is written as Mg/grid; should it be written as Mg/grid/yr.

Figure 2 Figure title indicates that fractional changes are shown. Actually percent changes are shown.

Figure S2 It contains four captions: (a-d). However, (c) is written twice, one should be written as (d).

---

## Referee Comment (RC2) · Anonymous Referee #1 · 4 Dec 2017

This manuscript presents an updated inventory of HCl and Cl2 emissions from various sources in China. Sources of inorganic chlorine are - in general - poorly constrained, so this study is highly relevant and provides new information on both the sources and the impacts of chlorine as atmospheric oxidant. I suggest it is published in ACP, after the authors have addressed a few questions and clarifications.

On page 5, line 4: what is the rationale behind this assumption? Also, is all coal used in China from domestic sources or is some of it imported? In the latter case, is the chlorine content different?

In Section 2.2, emissions from prescribed waste incineration are described. What

about the open waste incineration?

In Section 2.3, HCl is emitted primarily from the industrial sector. Is this due mostly to coal burning and if so for what purpose? I assume electricity generation is not included in this, but in the power plant sector.

page 12, line 26: do you mean "the highest chlorine emissions"? And do you mean Cl atoms or HCl and Cl2?

MINOR CORRECTIONS:

reactions 1 and 2: the 'H' should be lowercase

page 9, line 21: "literature"

page 10, line 22: what is "fine Cl-"?

In Table 1, maybe highlight the "Mainland China" line so that it is clear it is the sum of the previous lines?

In Figure 2, is "waste incineration" only the prescribed or the sum of prescribed and open?

In the Supplement, Figure S2 is split between two pages.

---

## Author Comment (AC1) · 5 Jan 2018

Dear Editor and Reviewers,

We would like to thank Referee #2 for valuable comments on our manuscript. We have careful revised our manuscript according to the referee's comments. We have addressed the referee's comments and concerns point by point as follows (comments from referee in black, author's response in red and author's changes in the revised manuscript in blue):

Comment:

The authors estimated chlorine emissions from coal combustion and prescribed waste incineration in China and used the CMAQ model to examine the impact of these chlorine emissions on ozone. Overall, the article is written clearly and merits publication. However, several issues need to be addressed before publication.

Responses:

We thank the referee for his nice comments.

Changes in the manuscript:

No changes were needed for this comment.

Comment:

Page 2, line 16-17, equation 1-2 The authors may replace "H" in both reactions with "h".

Responses:

We have corrected the typo.

Changes in the manuscript:

We have corrected it in line 16-17 on page 2 in the revised manuscript.

Comment:

Page 6, line 19-21 Incomplete sentence

Responses:

We have revised this sentence to "Table 1 also lists the waste incineration from garbage disposal incinerators in each province/city from China Urban-Rural Construction Statistical Yearbook".

Changes in the manuscript:

We have corrected it in line 26-28 on page 6 in the revised manuscript.

Comment:

Page 7, line 19-29 How the chlorine emissions were temporally allocated?

Responses:

To include the ACEIC in the CMAQ model, the chlorine emissions from different economic sectors were temporally allocated in different ways. For the coal combustion from the power plant, industrial and residential sectors, we distributed the total chlorine emissions into each month according to Wu (2009). The monthly variations of chlorine emissions from each sector are shown in Fig. R1. In addition, the daily distributions of chlorine emissions from the power plant, industrial and residential sectors were allocated the same way as the allocations of the MIX inventory from the corresponding sectors developed by Tsinghua University (http://www.meicmodel.org). For the coal combustion from other sector, the total chlorine emission was divided equally into each month, each day and each hour.

Since the burning process of garbage disposal incinerators is similar to that of the power plants, we assumed the same monthly and daily variation of prescribed waste incineration as that of the power plant sector.

[Figure]

Figure R1 Monthly distributions of chlorine emissions from the power plant, industrial, residential and other sectors.

References:
Wu, X. L.: The study of air pollution emission inventory in Yangtze Delta, M.S. thesis, Fudan University, China, 94 pp., 2009.

Changes in the manuscript:
The above statements have been added in line 22-29 on page 9 in the revised manuscript. We also added the reference (Wu, 2009) in line 22-23 on page 18.

Comment:
Page 9, line 13-15 ACEIC has been defined before. It appears that ACEIC has been defined several times throughout the article. Please check and define it once.
Responses:
We have corrected this redundancy. It should read only once now.
Changes in the manuscript:
We have corrected it in line 20-21 on page 9 and line 14-15 on page 13 in the revised manuscript.

Comment:
Page 11, line 1-10 The authors reported that chlorine emissions/chemistry increased the conversion of $NH_3$ into $NH_4^+$. What mechanism caused the increased conversion of $NH_3$ into $NH_4^+$?
Responses:
Volatile acidic species (i.e., HCl) can be partitioned into particles by neutralization reactions (Seinfeld and Pandis, 1998). Semi-volatile $NH_4Cl$ is formed via reversible phase equilibrium with $NH_3$ and HCl (Pio and Harrison, 1987). When the HCl emission was included in the model, HCl reacts with $NH_3$ to produce particulate $NH_4^+$ and $Cl^-$, provided that the $NH_3$ emission was sufficiently high.

References:
Pio, C. A., and Harrison, R. M.: Vapour pressure of ammonium chloride aerosol: effect of

temperature and humidity, Atmos. Environ., 21(12), 2711–2715, doi:10.1016/0004-6981(87)90203-4, 1987.

Seinfeld, J. H., and Pandis, S. N.: Atmospheric Chemistry and Physics, John Wiley & Sons, Inc., New York, 1998.

Changes in the manuscript:

The above statements have been added in line 12-21 on page 11 in the revised manuscript.

Comment:

Page 12, line 9-21 The authors demonstrated the impact of chlorine emissions/chemistry on daily maximum 1-hr $O_3$. Chlorine emissions/chemistry tends to increase ozone in the morning hours. Thus, it is likely to have a larger impact on 8-hr $O_3$ on than on 1-hr $O_3$. It will be important to present impact of chlorine emissions/chemistry on 8-hr $O_3$ also.

Responses:

Figure R2 presents the impact of chlorine emissions on 8-h $O_3$. The inclusion of chlorine emission in the CMAQ model increased the monthly mean daily maximum 8-h $O_3$ by up to 2.0 ppbv (4.1%), similar to the impact of chlorine emission on 1-h $O_3$. It is reasonable that 8-h $O_3$ is more representative than 1-h $O_3$. Hence, we moved the results of 1-h $O_3$ to the supplementary (Fig. S9a-c) and included that of 8-h $O_3$ in the manuscript (Fig. 8a-c).

| Base | Base-NoCl | (Base-NoCl)/NoCl×100% |
|------|-----------|------------------------|

[Figure]

Figure R2 Comparison of the monthly mean concentrations of daily maximum 8-h $O_3$ in the Base experiment, the differences (Base minus NoCl), and the percent changes to the NoCl experiment.

Changes in the manuscript:

The above statements have been added in line 1-5 on page 13 in the revised manuscript. We placed Fig. R2a-c (the results of 8-h $O_3$) to Fig. 8a-c in the revised manuscript and moved Fig. 8d-f (the results of 1-h $O_3$) in the original manuscript to the supplementary as Fig. S9a-c. Besides, the results of 1-h $O_3$ in Abstract (line 27 on page 1) and Conclusions (line 27-28 on page 13) were replaced with that of 8-h $O_3$.

Comment:

Page 13, line 1-10 The authors reported that chlorine emissions/chemistry decreased $NO_x$ concentrations without providing any explanation? Why $NO_x$ concentration decreased?

Responses:

The increase of Cl radical concentration can enhance the atmospheric oxidation, leading to the

increase of ozone and OH radical concentrations. The OH radicals can in turn oxidize $NO_x$ to produce particulate $NO_3^-$, resulting in the slight decrease of $NO_x$ concentration.

Changes in the manuscript:

The above statements have been added in line 29-31 on page 12 and line 1-10 on page 13 in the revised manuscript. Also, we placed Fig. 8a-c (the results of $NO_x$) in the original manuscript to Fig. 8d-f in the revised manuscript.

Comment:

Figure 1 Unit of chlorine emissions is written as Mg/grid; should it be written as Mg/grid/yr.

Responses:

We have revised "Mg grid$^{-1}$" to "Mg grid$^{-1}$ yr$^{-1}$" in Figure 1, S1, S2 and S3.

Changes in manuscript:

We have corrected it in Fig. 1 in the revised manuscript and Fig. S1, S2 and S3 in the revised supplementary.

Comment:

Figure 2 Figure title indicates that fractional changes are shown. Actually percent changes are shown.

Responses:

We have revised "fraction change" to "percent changes" in Fig. 4 and 8.

Changes in manuscript:

We have corrected it in Fig. 4 and 8 in the revised manuscript.

Comment:

Figure S2 It contains four captions: (a-d). However, (c) is written twice, one should be written as (d).

Responses:

We have corrected this typo.

Changes in the manuscript:

We have corrected it in Fig. S2 in the revised supplementary.

---

## Author Comment (AC2) · 5 Jan 2018

Dear Editor and Reviewers,

We would like to thank Referee #1 for valuable comments on our manuscript. We have careful revised our manuscript according to the referee's comments. We have addressed the referee's comments and concerns point by point as follows (comments from referee in black, author's response in red and author's changes in the revised manuscript in blue):

Comment:

This manuscript presents an updated inventory of HCl and $Cl_2$ emissions from various sources in China. Sources of inorganic chlorine are - in general - poorly constrained, so this study is highly relevant and provides new information on both the sources and the impacts of chlorine as atmospheric oxidant. I suggest it is published in ACP, after the authors have addressed a few questions and clarifications.

Responses:

We thank the referee for nice comments.

Changes in the manuscript:

No changes were needed for this comment.

Comment:

On page 5, line 4: what is the rationale behind this assumption? Also, is all coal used in China from domestic sources or is some of it imported? In the latter case, is the chlorine content different?

Responses:

For those regions where chlorine contents are not listed in the literature, we estimated the chlorine emissions using the average chlorine content ($280 \ \mu g \ g^{-1}$) in China according to Chen (2010).

Some of the coals consumed in China were imported from other countries which might have different chlorine contents. According to the report of China Energy Statistical Yearbook (CESY, National Bureau of Statistics, 2013), the total amount of coals imported into China was 288 Tg and the total coal consumption in China was 3526 Tg in 2012. Over 91% of the coals were domestically produced in China. It is hence concluded that the different chlorine content of the imported coals has minor influence on the estimation of chlorine emission in China. However, it is difficult to evaluate to what extent the influence is. Hence, we estimated the chlorine emission from coal combustion in China using the chlorine content of coal from domestic sources and did not take the different chlorine content of the imported coals into account.

References:

Chen, L. H.: Study on environmental geochemistry of chlorine in Chinese coals, M.S. thesis, Nanchang University, China, 46 pp., 2010.

National Bureau of Statistics: China Energy Statistical Yearbook 2013, China Statistics Press, Beijing, 2013.

Changes in the manuscript:

The above statements have been added in line 2-12 on page 5 in the revised manuscript.

Comment:

In Section 2.2, emissions from prescribed waste incineration are described. What about the open waste incineration?

Responses:

Open waste incineration is the uncontrolled emissions from both residential and dump waste burning. In this study, as the article title mentioned, we only focus on the anthropogenic chlorine emissions from coal combustion and prescribed waste incineration.

Changes in the manuscript:

No changes were needed for this comment.

Comment:

In Section 2.3, HCl is emitted primarily from the industrial sector. Is this due mostly to coal burning and if so for what purpose? I assume electricity generation is not included in this, but in the power plant sector.

Responses:

HCl is emitted primarily from the industrial sector, due mostly to the coal burning. The referee is correct that electricity generation is not included in the industrial sector but in the power plant sector. Many industrial processes (e.g., iron and steel processing, non-ferrous metals processing, cement production) that need coal burning are included in the industrial sector, leading to the highest source of HCl.

Changes in the manuscript:

The above statements have been added in line 6-7 on page 8 in the revised manuscript.

Comment:

page 12, line 26: do you mean "the highest chlorine emissions"? And do you mean Cl atoms or HCl and $Cl_2$?

Responses:

Yes, it means the highest chlorine emissions and it means HCl and $Cl_2$. We have revised this description to "the highest emissions of HCl and $Cl_2$".

Changes in the manuscript:

We have clarified this description in line 16-17 on page 13 in the revised manuscript.

Comment:

reactions 1 and 2: the 'H' should be lowercase

Responses:

We have corrected this typo.

Changes in the manuscript:

We have corrected it in line 16-17 on page 2 in the revised manuscript.

Comment:

page 9, line 21: "literature"

Responses:

We have corrected this typo.

Changes in the manuscript:

We have corrected it in line 3 on page 10 in the revised manuscript.

Comment:

page 10, line 22: what is "fine Cl⁻"?

Responses:

It means fine particulate $Cl^-$. It has been revised to clarify this point.

Changes in the manuscript:

We have clarified it in line 6 on page 11 in the revised manuscript.

Comment:

In Table 1, maybe highlight the "Mainland China" line so that it is clear it is the sum of the previous lines?

Responses:

We have highlighted the "Mainland China" line with black border.

Changes in the manuscript:

We have marked this line in Table 1.

Comment:

In Figure 2, is "waste incineration" only the prescribed or the sum of prescribed and open?

Responses:

"Waste incineration" in Fig. 2 was only the prescribed waste incineration. It has been revised to clarify this point.

Changes in the manuscript:

We have clarified it in Fig. 2.

Comment:

In the Supplement, Figure S2 is split between two pages.

Responses:

We have corrected this issue.

Changes in the manuscript:

We have corrected it in Fig. S2.